# A computationally light-weight model for ensemble forecasting of environmental hazard: General TAMSAT-ALERT v1.2.1

Emily Black[1,2], John Ellis[3], Ross I. Maidment[1,2,4]

[1]Department of Meteorology, University of Reading, Reading, RG6 6BB, UK
5   [2]National Centre for Atmospheric Science, Leeds, LS2 9PH, UK
[3]Department of Computer Science, University of Warwick, CV4 7AL, UK
[4]National Centre for Earth Observation, Leicester, LE4 5SP, UK

*Correspondence to*: Emily Black (e.c.l.black@reading.ac.uk)

## Abstract

Efficient methods for predicting weather-related hazards are crucial for the effective management of environmental risk. Many environmental hazards depend on the evolution of meteorological conditions over protracted periods, requiring assessments that account for evolving conditions. The TAMSAT-ALERT approach addresses this challenge by combining observational monitoring with a weighted multi-year ensemble. As such, it enhances the utility of existing systems by enabling users to combine multiple streams of monitoring and meteorological forecasting data into holistic hazard assessments. TAMSAT-ALERT forecasts are now used in a number of regions in the Global South for soil moisture forecasting, drought early warning and agricultural decision support. The model presented here, General TAMSAT-ALERT, represents a significant scientific and functional advance on previous implementations. Notably, General TAMSAT-ALERT is applicable to any variable for which time series data are available. In addition, functionality has been introduced to account for climatological non-stationarity (for example due to climate change); large-scale modes of variability (for example El Nino), and persistence (for example of land-surface condition). In this paper, we present a full description of the model, along with case studies of its application to prediction of Central England Temperature, Pakistan vegetation condition and African precipitation.

**Short summary**

We present General TAMSAT-ALERT: a computationally lightweight and versatile tool for generating ensemble forecasts from time series data. General TAMSAT-ALERT is capable of combining multiple streams of monitoring and meteorological forecasting data into probabilistic hazard assessments. As such, it complements existing systems and enhances their utility for actionable hazard assessment.

## 1    Introduction

Transparent, robust and computationally efficient methods for hazard assessment are of great value to stakeholders dealing with environmental risk (for example, Boult et al. 2022). Weather-related hazards may depend on the evolution of multiple meteorological variables over a protracted period. For example, crop yield is affected by precipitation and temperature throughout the growing season. In-season updates therefore require monitoring of past conditions as well as forecasting. Combining observations and forecasts is, however, challenging - especially when the hazard in question is affected by more than one variable. Extending the aforementioned example: when making an in-season assessment of the risk of low crop yield, it is necessary to consider both meteorological conditions in the period since planting, and the probability of adverse conditions during the remaining season. One approach would be to drive a crop model, such as AquaCrop (Steduto et al. 2009), with observations of each driving variable up until the present, and with an ensemble of numerical weather prediction (NWP) model

forecasts for the future. There are two difficulties with this. On a practical level, it may be problematic to obtain forecasts of all driving variables at the required time and spatial resolution. A further challenging problem is the drift in predictions that occurs as NWP models move from their initial state into equilibrium with their own physics (for example, Manzanas 2020). If model predictions were to be spliced directly onto historical observations, the drift would cause systematic bias in yield assessments, the magnitude of which would depend on the stage of the growing season at which the meteorological forecasts were initiated. This is a challenging error to correct using standard bias correction methods, because the magnitude of the bias depends on the lead time of the forecast. The TAMSAT-ALERT approach addresses these issues by splicing together historical time series for the past with an ensemble comprised of time series extracted from individual historical years (hereafter referred to as a 'climatology') for the future (Asfaw et al. 2018, Boult et al. 2020, Brown et al. 2017). The use of consistent historical data avoids the issue of model drift, and accounts for complex interactions between variables, such as precipitation, temperature and net radiation (Asfaw et al. 2018). From a practical perspective, seamless integration of past and future conditions facilitates in-season updates for slowly developing hazards, such as drought (Brown et al. 2017). This feature of TAMSAT-ALERT provides a simple means of calculating ensemble statistics that take into account both historical time series and future predictions. Furthermore, the progressive incorporation of observational monitoring gradually reduces uncertainty in risk assessments as the season evolves - facilitating decision-making.

In its default state, TAMSAT-ALERT treats all members of the climatological ensemble as equally likely. When calculating ensemble statistics, such as ensemble mean and standard deviation it is, however, in principle possible to weight ensemble members individually, if there is evidence that some are more likely than others. For example, climatological ensemble members may be weighted more strongly if the El Nino phase during their associated year is close to that at the initiation of the forecast. Extending this idea, the ensemble can be weighted using meteorological forecasts, based on the similarity of the predicted weather to the weather during the year associated with each ensemble member. A strength of the methodology is that NWP output can be incorporated, even when forecasts are not available for the variable being assessed. In Kenya, for example, incorporation of skilful precipitation tercile forecast probabilities output by the ECMWF dynamical forecasting system improves the skill of NDVI and yield forecasts during the secondary rainy season (Young et al. 2020, Boult et al. 2020). The use of a weighted climatological ensemble thus enables historical and forecast data to be combined into a holistic view of risk (see Appendices A and B for more comprehensive description of the weighting methodology).

Previous work on the TAMSAT-ALERT method has described how the system can be used for agricultural and drought forecasting (Asfaw et al. 2018; Boult et al. 2020, Black et al. 2024) and to short term decision support (Black et al. 2023). Although originally designed for application in Africa, there have been applications in Guyana (David 2023) and Pakistan (Black et al. 2024). A similar approach is used for the FEWSNET drought outlooks (Shukla et al. 2014, Turner et al. 2022) and for precipitation predictions within the seasonal performance probability tool (Novella and Thiaw 2016).

In this paper, we present General TAMSAT-ALERT - a versatile implementation of the TAMSAT-ALERT framework. Unlike the previous system, which required the use of a land-surface or crop model, General TAMSAT-ALERT, can be applied to any variable for which time series data are available. General TAMSAT-ALERT is thus a self-contained model, rather than a modelling framework. During the development of General TAMSAT-ALERT, we took the opportunity to extend the methodology and completely revise the way that users interact with the code. A key innovation is the option to increment variables from the forecast initialisation date, enabling forecasts to account for persistence of environmental conditions in time. In addition, as well as enabling users to weight the ensemble with time series of climate indices, General TAMSAT-ALERT permits weighting by proximity of the climatological year to the target year. The latter development avoids the assumption of climatological stationarity - a weakness of the original methodology.

The paper is structured as follows. Section 2 summarises the design and implementation of General TAMSAT-ALERT and describes the novel developments in TAMSAT-ALERT. In Section 3, the usage of the system is illustrated through three case studies: probabilistic prediction of Central England Temperature statistics; NDVI forecasting in Pakistan; and continually updated predictions of the standardised precipitation index (SPI) for Africa. The paper closes with some reflections on how General TAMSAT-ALERT fits into the ever-expanding ecosystem of environmental forecast models.

## 2    Model description

### 2.1    Conceptual approach and implementation

The inputs and outputs of the system are illustrated by Figure 1 and the procedure is described more fully in the following example, which considers a metric calculated over a period of interest, with the forecast initiated on a date within the period of interest.

Preparatory steps:

1. Specify the period of interest (POI) start and end dates, the initiation date of the forecast, and supply the time series data for the forecasts.
2. Determine the dominant periodicity at the time resolution of the data (see Appendix C). For example, for a time series with an annual dominant periodicity, the periodicity would be 365 for daily data or 12 for monthly data). To simplify the description, for the rest of this section, we will assume that the dominant periodicity is annual.
3. Optionally, transform the time series into a time series of incremental change (i.e. differences between each value and the one that came before it in the time series)
4. Divide the time series into individual years

The forecast:

5. Extract the data value on the initiation date of the forecast
6. Extract the future forecast period from every member of the climatological ensemble. The first day of the forecast period is the initiation date, and the last day is the end of the period of interest.
7. If the option is taken to increment from the initiation date, the data derived in step 5 is added to the state on the initiation date of the forecast, otherwise the ensemble consists of the raw data derived in step 5.

Calculation of the metric over the whole period of interest:

8. The period of interest spans a period starting prior to the initiation date as well as a period afterwards. For each ensemble member, the forecast ensemble member (derived in step 6) is spliced together with the observations from the start of the period of interest to the initiation date

Calculation of ensemble statistics:

9. Any weighting of the ensemble is applied at this stage, with weights allocated to each climatological ensemble member, based on the conditions experienced during the year from which the ensemble member is derived (see Appendix A and B for further details of the methodology). The code includes three options for weighting:
    a. No weighting applied (weighting flag set to 0). In this case, equal weights are assigned to all ensemble members
    b. Weighting by proximity of the climatological ensemble member year to the actual year (weighting flag set to 1). This option allows the user to account for long term variability and trends in the variable being forecast. The strength of the weighting is assigned by the user.

c. Weighting using an index of climate variability, such as ENSO. This option allows the user to favour climatological ensemble members in which climate conditions are similar to those at the initialisation of the forecast. Both the data file used and the strength of the weighting are specified by the user.

It is recommended that users experiment with different weightings and strengths, ideally carrying out skill assessments for a range of different set ups (see case study 1 for an example of this).


10. Any ensemble forecast statistic can be calculated from the ensemble derived in step 7, using any weighting options derived in step 8. These include ensemble mean, ensemble spread, confidence intervals and probability that the metric being forecast will be less than a user-prescribed threshold. The in-built returns from the General TAMSAT-ALERT are weighted ensemble mean and weighted ensemble standard deviation.


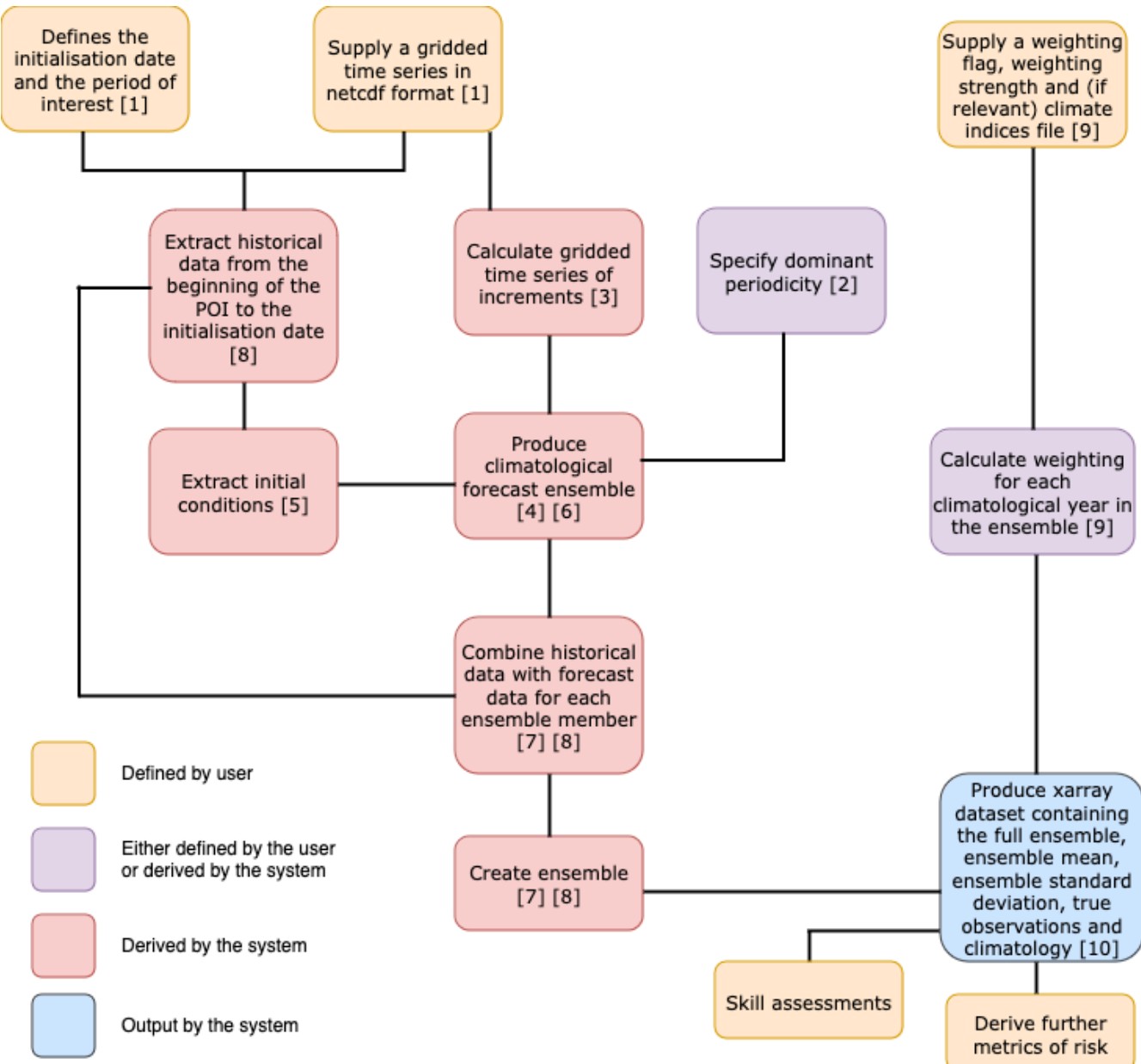

**Figure 1: Flow chart showing how the general TAMSAT-ALERT implementation works. The numbers in square brackets refer to the modelling steps described in Section 2.1**

## 2.2 Novel methodological developments

The original TAMSAT-ALERT implementation was designed as a framework for producing ensemble crop yield forecasts (Asfaw et al. 2018). Following the publication of the original TAMSAT-ALERT method, there was interest from the agricultural and humanitarian sectors in wider application to drought forecasting and agricultural decision support (Boult et al. 2020, Black et al. 2023). These subsequent applications did not require significant development of the underlying methodology for the framework, and so there was no need to develop a new model. The wider use of the TAMSAT-ALERT

methodology, however, highlighted an opportunity to develop the methodological framework described in these papers into a new, more general model. General TAMSAT-ALERT thus builds on the success of TAMSAT-ALERT by enabling users to derive forecasts directly from observations and reanalysis, without the need for the use of land-surface/crop models or NWP forecasts. This has required several methodological and scientific extensions, and a complete redesign and rewrite of the underlying code. For this release of General TAMSAT-ALERT, we have also taken the opportunity to rethink the way that

users interact with the system, with the aim of enabling a wider range of users to use the system to produce quantitative forecasts.

Scientific developments:

- In the previous implementation of TAMSAT-ALERT, there was an assumption of climatological stationarity, which is clearly over simplistic when analysing variables with significant trends (such as surface temperature). General

TAMSAT-ALERT includes options to account for trends in variables by incrementation from the last day of the historical time series and/or by weighting the climatology to favour years closer to the forecast/hindcast initiation date. These methodologies are demonstrated in Case Study 1: prediction of Central England Temperature extremes

- The previous implementation of TAMSAT-ALERT did not explicitly account for the persistence, instead relying on land-surface models to represent persistence in soil moisture. General TAMSAT-ALERT directly accounts for

temporal persistence (and implicitly long term trends) by allowing the user to select an option for incrementing forecasts from the last day of historical observations. This functionality is illustrated by Case Study 2: NDVI forecasting for Pakistan and by Case Study 1: prediction of Central England Temperature extremes

- Predictability of many environmental variables is amplified by large scale modes of variability, such as ENSO. In General TAMSAT-ALERT, a method is implemented for incorporating climate indices into ensemble predictions.

This is illustrated by Case Study 3: probabilistic prediction of SPI3 for Africa.

Methodological developments:

- General functions for weighting ensembles with climatological indices have been developed and incorporated into the code (allowing the user to specify the strength of the weighting). Further details are included in Appendix B.
- An extension has been included to identify the dominant periodicity in input data, and the code has been generalised

to detail with data with any periodicity and time resolution. Further details are included in Appendix C.

Code developments

- General TAMSAT-ALERT is released as fully documented and publicly issued python package (general_tamsat_alert)
- The whole procedure described in Figure 1 and Section 2.1 is encompassed by a single function

[`do_forecast()`], which ingests netcdf data, and produces ensemble forecasts for a user specified initiation date and output period. The function includes all of the functionality for weighting and incrementing described above.

## 3 Model application

The application of General TAMSAT-ALERT is demonstrated through three case studies, designed to demonstrate the functionality of the system beyond conventional analyses of meteorological time series data. In particular, the case studies demonstrate how General TAMSAT-ALERT improves skill by incorporating non-stationarity, persistence of environmental variables in time, and wider modes of variability, such as ENSO.

- Case study 1: prediction of Central England Temperature (CET) extremes There is a strong anthropogenic positive trend in CET, superposed on strong decadal and interannual variability. Application of General TAMSAT-ALERT to prediction of CET demonstrates how the system handles non-stationarity in historical climate

- Case study 2: prediction of NDVI in Pakistan NDVI in semi-arid regions is highly persistent and so this case study demonstrates how the General TAMSAT-ALERT exploits persistence to make skilful predictions with lead times of up to two months

- Case study 3: probabilistic prediction of the three-month standardised precipitation index (SPI3) for Africa. SPI3 on a particular day depends on cumulative precipitation in the preceding three months. This case study demonstrates how the system combines observations and predictions into probabilistic forecasts. September-November precipitation in East Africa is, furthermore, strongly influenced by El Nino, and the case study demonstrates how weighting with the Oceanic Nino Index (ONI) improves forecast skill

In order to assess the skill of the forecasts objectively, two standard skill scores were utilised in the case studies:

- $r^2$ (square of the Pearson's moment correlation coefficient) is a measure of how closely the hindcast interannual time series of NDVI captures the observed time series. Specifically, $r^2$ gives the proportion of variance in the time series of observations is explained by the hindcasts. An $r^2$ of 1 denotes a perfect forecast, in the observed time series is replicated precisely by the hindcasts; for a 40-year time series, $r^2 < 0.07$ ($r < 0.26$) denotes that the correlation is insignificant at the 95% level.

- ROC-AUC (Receiver Operating Characteristics – Area Under Curve) is a metric of how reliably events can be predicted. In this context, an event is the predictand being below a user-defined threshold (for example, below the 20th percentile). An ROC-AUC of 0.5 or less indicates no skill in distinguishing false alarms from true positives (hits). An ROC-AUC of 1 indicates a perfect forecast. For a concise guide to ROC-AUC, see https://www.metoffice.gov.uk/research/climate/seasonal-to-decadal/gpc-outlooks/user-guide/interpret-roc

### 3.1 Case study 1: Prediction of Central England Temperature (CET) seasonal extremes

#### 3.1.1 Case study 1: Introduction

In this case study, the TAMSAT-ALERT system provides probabilistic predictions of whether or not CET in a given season will exceed the 90th, 95th or 99th percentiles. CET exhibits a pronounced trend, which is reflected in increased occurrence of 90th, 95th and 99th percentile events in the later part of the time series (Figure 2). The TAMSAT-ALERT system can account for non-stationarity in two ways. Firstly, for future periods, the climatological ensemble can be incremented from the conditions on the initiation date of the forecast. Any trend in the variable being predicted on the initiation date will be implicitly represented in the ensemble. Secondly, the facility to weight the climatological ensemble allows the system to favour years that are close in time to the period of interest (see Appendix A and B). The CET case study thus illustrates how accounting for non-stationarity improves prediction skill for variables with a strong trend.

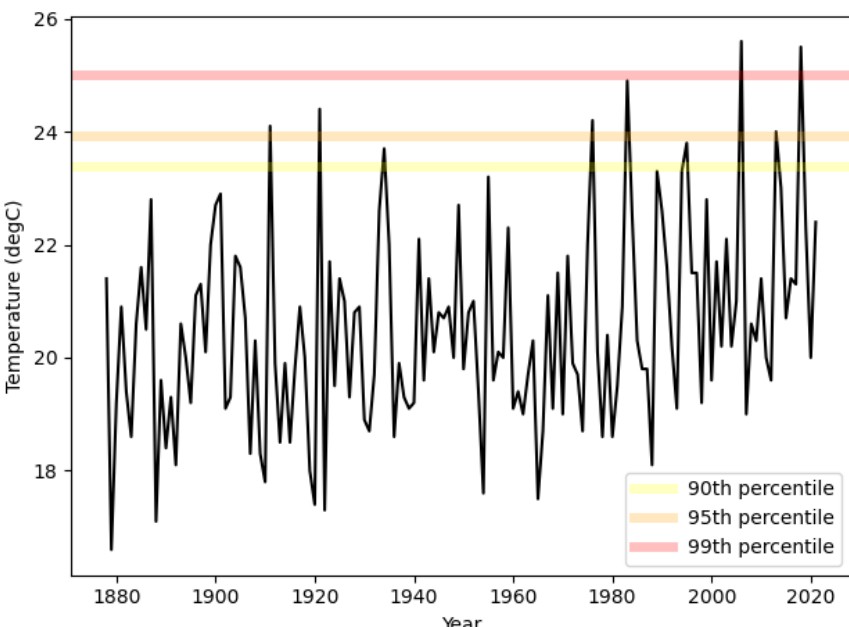

**Figure 2: Time series of monthly mean daily maximum July Central England Temperature (CET), highlighting the 90th, 95th and 99th percentile**

### 3.1.2 Case study 1: Data and methodology

The dataset used in this study is HadCET (available from https://www.metoffice.gov.uk/hadobs/hadcet/). HadCET is a time series of temperature diagnostics for a region of the UK, roughly encompassed by a triangle enclosed by Lancashire, London

and Bristol. Monthly CET data are provided from 1659, with daily maximum and minimum temperatures available from 1878 (see Parker et al. 1992 for a full description). This case study utilises the monthly mean daily maximum temperatures for 1882-2021, focusing on July. It can be seen from the time series displayed in Figure 2 that there is pronounced decadal variability throughout the time series, with a clear warming trend evident from ~1980 onwards.

Hindcasts were generated for the whole data period, and each hindcast uses data from all years to generate the climatological

ensemble (i.e. from the past as well as from the present). The metrics derived from the system were probabilistic forecasts that the 90th, 95th and 99th percentiles of CET will be exceeded. The percentiles were calculated using data from the whole time series, based on the ensemble mean and standard deviation (i.e. assuming Gaussian behaviour). As was described above, General TAMSAT-ALERT can optionally be run with each ensemble member incremented from the initiation dates of the forecasts and/or weighted according to some measure of how similar the climatological year is to the year in question. The

intention of the incrementation process is to generate ensembles of increments of the historical time series, relative to the previous time step. So, say for a given climatological year July had an average temperature of $18^0$C and August had an average temperature of $17.5^0$C. In this case the incremental value for August would be $-0.5^0$C. The climatological ensemble member time series thus comprises a time series of increments, which are then applied from the initialisation date. Practically speaking, the increment time series are generated by subtracting the initial state for the climatological year and then adding the result to

the current year.

In the case study the system was run for four set ups. All set ups are run for a 1-month lead forecast (i.e. initialised by the June monthly mean):

- No incrementation or weighting (i.e. the ensemble of values for historical data)
- Ensemble members weighted by the proximity of the ensemble climatological year to the hindcast year but no

incrementation. In this case the exponential weighting factor is arbitrarily set to 1 (see Appendix B)

- No weighting but ensemble members incremented from the temperature on the time of initialisation
- Ensemble members weighted by the proximity of the ensemble climatological year to the hindcast year, and ensemble members incremented from the temperature on the day of initialisation

### 3.1.3 Case study 1: Results

Figure 3 shows forecasts for each of the setups described above. It can be seen that monthly mean daily maximum temperature for the example month chosen (July 2021) fell just below the 90th percentile. When no weighting or incrementing are included, the probabilities of exceeding the 90th, 95th and 99th percentile are close to the climatological expectation of 0.1, 0.05 and 0.01 respectively. When the weighting and/or incrementing is applied, the probabilities of exceeding the threshold increase markedly. This is, in part, because both weighting and incrementing implicitly account for the strong observed positive trend.


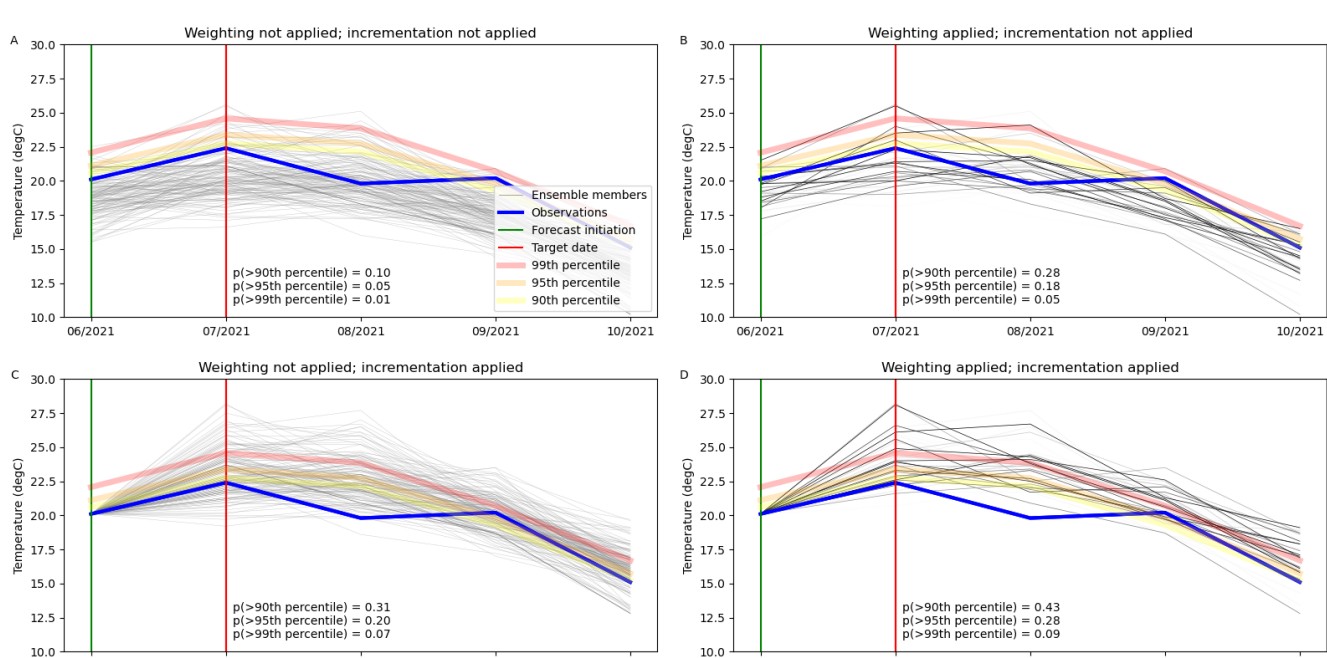

**Figure 3: Example forecasts with one-month lead for July 2021. A) No weighting or incrementation of the time series B) Weighting of the climatology by year proximity but no incrementation of the time series; C) Incrementation of the time series from the initiation but no weighting; D) Weighting by year proximity and incrementation of the time series. In all of the plots, observations are blue,**
**the 90th, 95th, and 99th percentiles are pale yellow, yellow and orange and individual ensemble members are grey. The green vertical line is the date of forecast initiation and the red line is the target date. In B and D, the weighting is denoted by the darkness of the lines.**

A more formal analysis is shown in Figure 4, which displays a time series of hindcast probabilities of exceeding the 90th, 95th and 99th percentiles. When neither incrementing nor weighting are applied (panel A), the probabilities are close to
climatological expectation and do not change from year to year. Weighting the distribution by proximity of the climatological ensemble year to the observed year (panel B) has the expected effect of reducing the probabilities in the early part of the time series and increasing them in the later part – reflecting the trend. The incrementing (panel C) has a similar effect, with the probabilities of exceeding each threshold being consistently elevated in the later part of the time series. Interestingly, when the incrementing is included, the probabilities are consistently higher than the climatological expectation – suggesting that
there is some degree of non-Gaussian behaviour that is picked up when the forecasts are initialised from observations. In this case, the elevated probabilities suggest that the climatological percentiles (as estimated by assuming Gaussian behaviour) are too cold. When both the incrementing and the weighting are applied (panel D), elements of both types of behaviour are evident, with strong increases in hindcast probabilities in the later part of the time series, and greater than climatological probabilities throughout. Figure 4 displays the results of a formal skill assessment (ROC-AUC). When neither incrementing nor weighting
are applied, the ROC-AUC is close to the climatological expectation of 0.5. When weighting and/or incrementing are applied,

the skill improves, with ROC-AUC in the region of 0.7-0.8. For the most extreme cases (95[th] and 99[th] percentile), slightly better scores were achieved when both the incrementing and weighting were applied, but the differences are small.

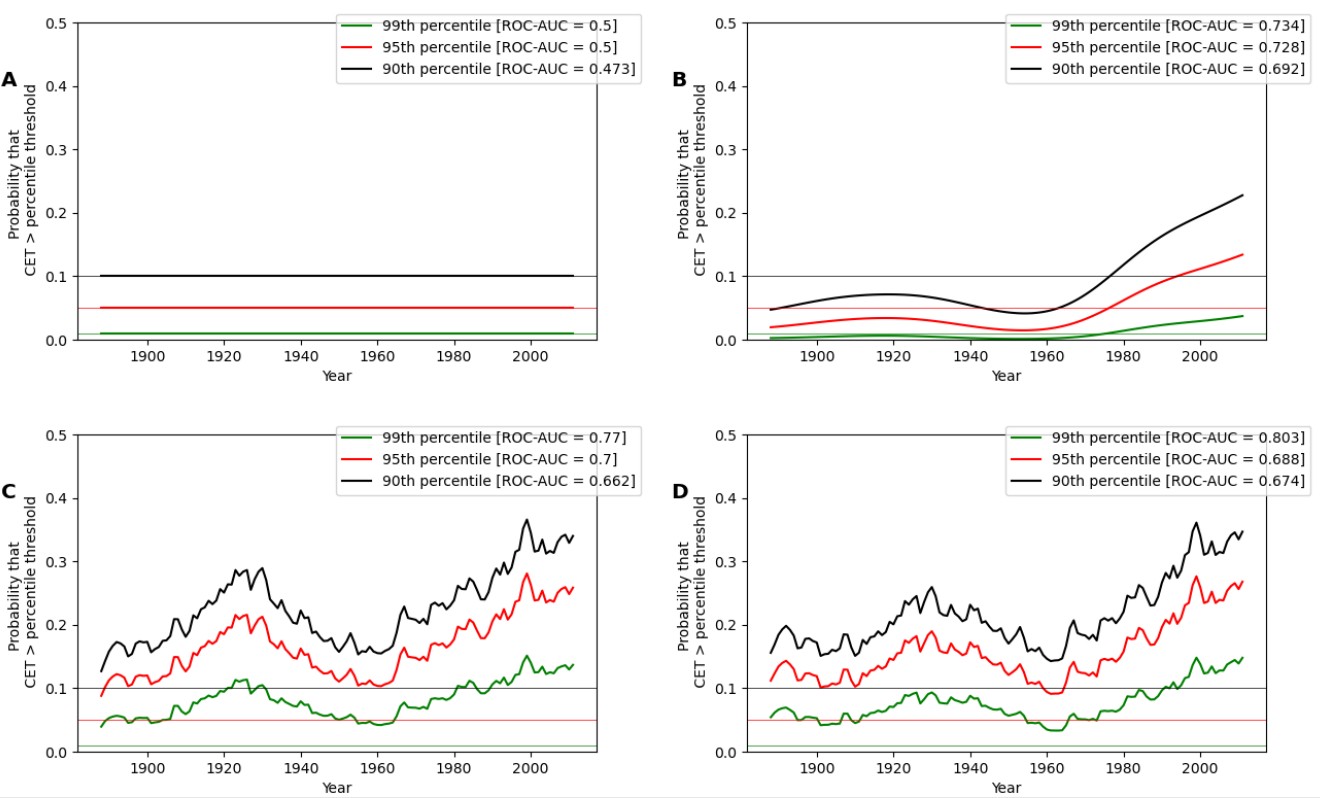

**Figure 4: Time series of predicted probabilities that the July Central England Temperature (CET) will exceed the 1850-2020 90[th], 95[th] and 99[th] percentile for a 1-month lead – i.e. forecasts initiated in June. A) No weighting or incrementation of the time series B) Weighting of the climatology by year proximity but no incrementation of the time series; C) Incrementation of the time series but no weighting; D) Both weighting and incrementation applied. The bold lines show modelled probabilities of exceedance of the 90[th], 95[th] and 99[th] percentile thresholds, and the fine lines show the climatological probabilities.**

### 3.2    Case study 2: Prediction of NDVI for Pakistan

The implementation of General TAMSAT-ALERT to NDVI forecasting is illustrated through a case study of Pakistan and the surrounding region. Pakistan was chosen for the case study for several reasons. Firstly, the climate and topography of Pakistan varies considerably (see Figure 5), enabling us to test the NDVI forecasting method in a range of environmental settings. The topographic zones include the highlands of the north, large river plains in Punjab and Sindh, and the Balochistan Plateau. The climate also varies considerably. For example, precipitation ranges from <100mm/year in the deserts of Balochistan to >1000mm/year in northern regions affected by the southwest monsoon. Secondly, vegetation in Pakistan is partly rainfed and partly fed by river overflow and glacial melt. This means that there is a disconnect between variability in precipitation and vegetation condition. Direct observation of vegetation may thus be the most appropriate method of monitoring drought and crop condition and NDVI forecasts thus have practical value for the region. The final reason for choosing Pakistan was pragmatic. In 2020, the START Network commissioned the TAMSAT group to develop a new drought monitoring service for Punjab and Sindh, which unlike pre-existing services, targets the secondary winter growing season. The development of the drought monitoring system necessitated extending the TAMSAT-ALERT method from soil moisture in Africa to NDVI in Pakistan (Black et al. 2024).

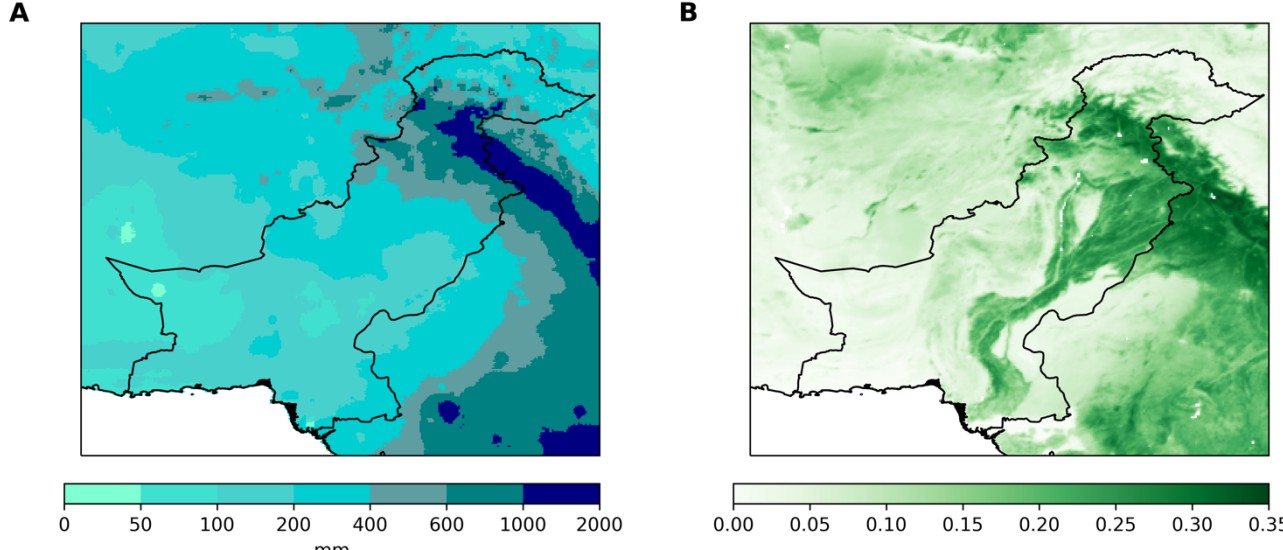

**Figure 5: Maps of precipitation and NDVI annual means. A) Annual mean precipitation in the case study region; B) Annual mean maximum 15-day NDVI in the case study region. The country outline shows the border of Pakistan.**

### 3.1.1 Case study 2: Data and methodology

The dataset used in this study is the Blended Vegetation Health Indices Product (Blended-VHP). Blended-VHP is a multiple-product dataset issued by NOAA's Center for Satellite Applications and Research (STAR) that produces global Vegetation Health products. The Blended-VHP products include data from different satellite sensors (VIIRS for 2013-present and AVHRR (1981-2012)), which have been re-processed into a single homogenous time series (Yang et al. 2021). The vegetation health-related variables provided, include NDVI, Brightness Temperature, Vegetation Condition Index, Thermal Condition Index and Vegetation Health Index. The raw Blended-VHP data are released as weekly files at 4km resolution. For this study, the data were re-gridded to $0.05^0$ and interpolated to twice monthly time steps (15th of the month, last day of the month). The forecasts were produced using incrementation but no weighting, with the TAMSAT-ALERT forecasts generated independently for each grid point.

### 3.1.2 Case study 2: Results

An example of an NDVI forecast for 1st August 2018, with lead time of up to 60 days is shown in Figure 6, and Figure 7 shows $r^2$ , together with ROC-AUC for forecasts of NDVI being below the 20th percentile on 1st August. The skill scores are calculated for each point for the gridded forecasts. Even two months in advance, there is some skill compared to climatology, and by 1st July (one month lead time), the forecasts are highly skilful over most of the region. Although there is some spatial variation, in all regions there is skill at a 60-day lead, with excellent skill at a 30-day lead (ROC-AUC > 0.85; $r^2$ > 0.75). Since no weighting is applied, and the forecasts are for an NDVI snapshot, rather than seasonal mean, the only source of skill in the forecasts is the time persistence of NDVI. As was discussed for CET, a potential limitation of the TAMSAT-ALERT method is the implicit assumption that we can predict the future using historical climatologies. The difficulty of dealing with data with a strong trend is partly addressed by deriving ensembles using time step changes in NDVI, rather than with the absolute values of the historical observations. It should be noted, however, that although starting the forecasts from the NDVI on the date of initiation accounts for the trend in NDVI on the date of initiation, it does not account for trends in the magnitude of the NDVI time increments.

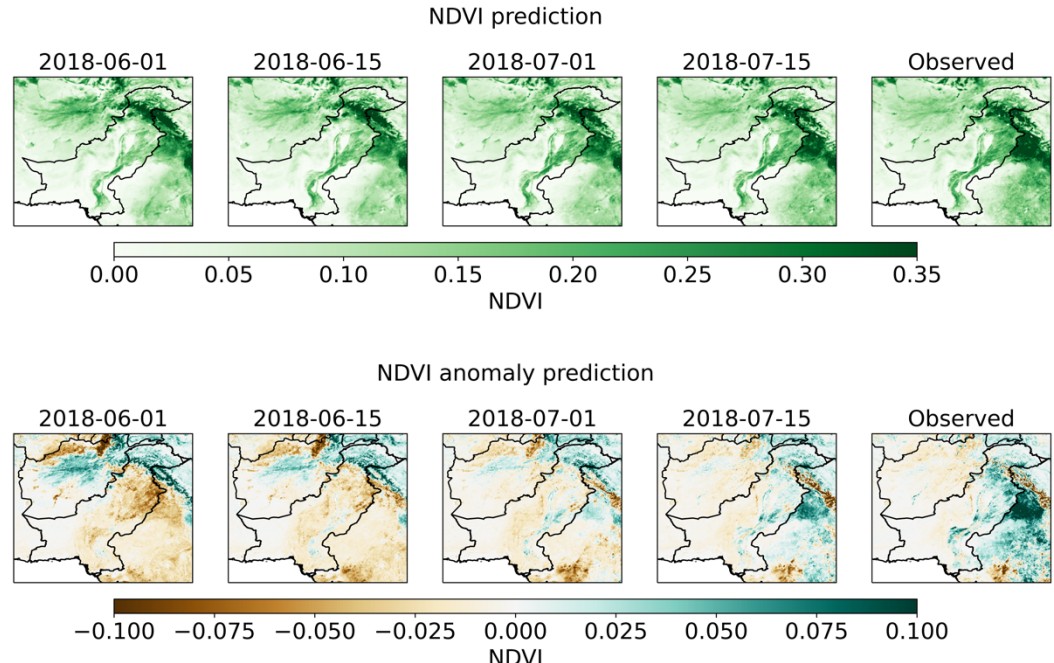

**Figure 6: Predicted and observed NDVI in an example month (August 2018). The dates on each sub-figure are the initialization dates for the forecasts. Top panel: predicted and observed NDVI; Bottom panel: predicted and observed NDVI anomaly**

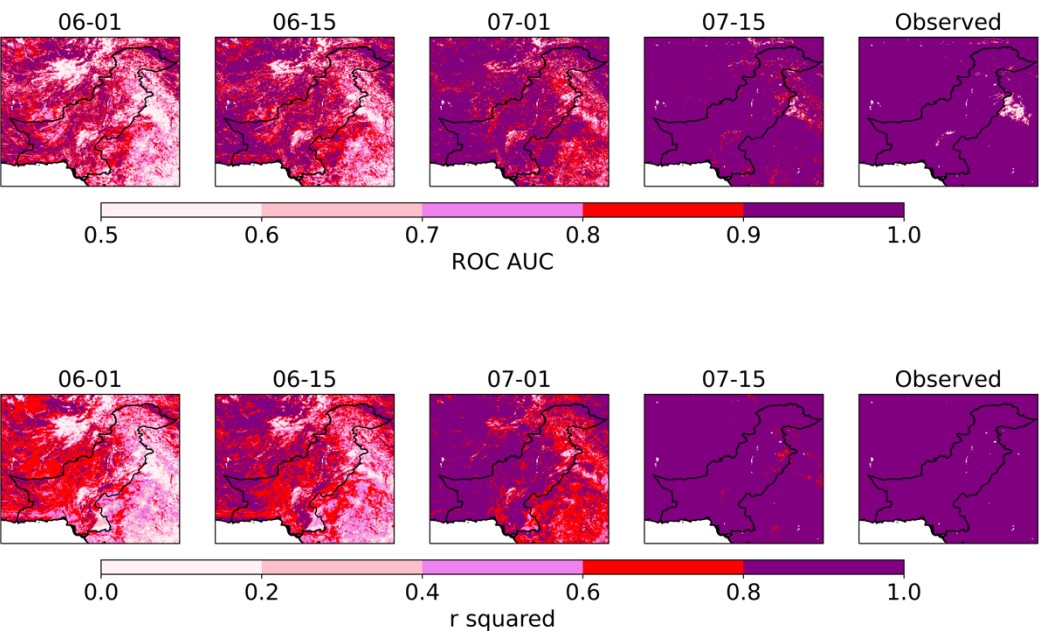

 **Figure 7: Skill scores for NDVI predicted for August 1st. The dates on each sub-figure are the initialization dates for the forecasts. Top panel: ROC AUC for detecting a 20th percentile event; Bottom panel: r-squared (i.e. the square of the Pearson correlation coefficient)**

### 3.3    Case study 3: 3-month Standardised Precipitation Index (SPI3) prediction for Africa

Lower than usual precipitation has severe consequences in Africa because of the region's dependence on rainfed agriculture.

 Precipitation in Africa varies strongly in both space and time, with most regions having pronounced dry and wet seasons (see Figure 8). A commonly used metric of precipitation anomaly is the Standardised Precipitation Index (SPI) (McKee et al. 1993).

For a given time, SPI is essentially the normalised and standardised cumulative precipitation anomaly, derived for a user-defined number of months preceding. The choice of number of months depends on how the SPI data will be utilised. For assessment of agricultural drought, a 3-month period is typically used; for hydrological drought, longer periods (6 or 9 months) may be more appropriate.

In this application, the TAMSAT-ALERT system is used to predict SPI3 for Africa with lead times of up to three months. As was described above, SPI3 is based on cumulative precipitation during the three months prior to the target date. At a three month lead, therefore, the TAMSAT-ALERT forecast is based entirely on an ensemble of future precipitation. As the season progresses, the ensemble progressively incorporates observations. For example, at a one-month lead, each ensemble member will include two months of observations and one month of forecasts. The primary aim of this case study is thus to demonstrate how TAMSAT-ALERT can be used to combine historical observations and climatological information into probabilistic predictions.

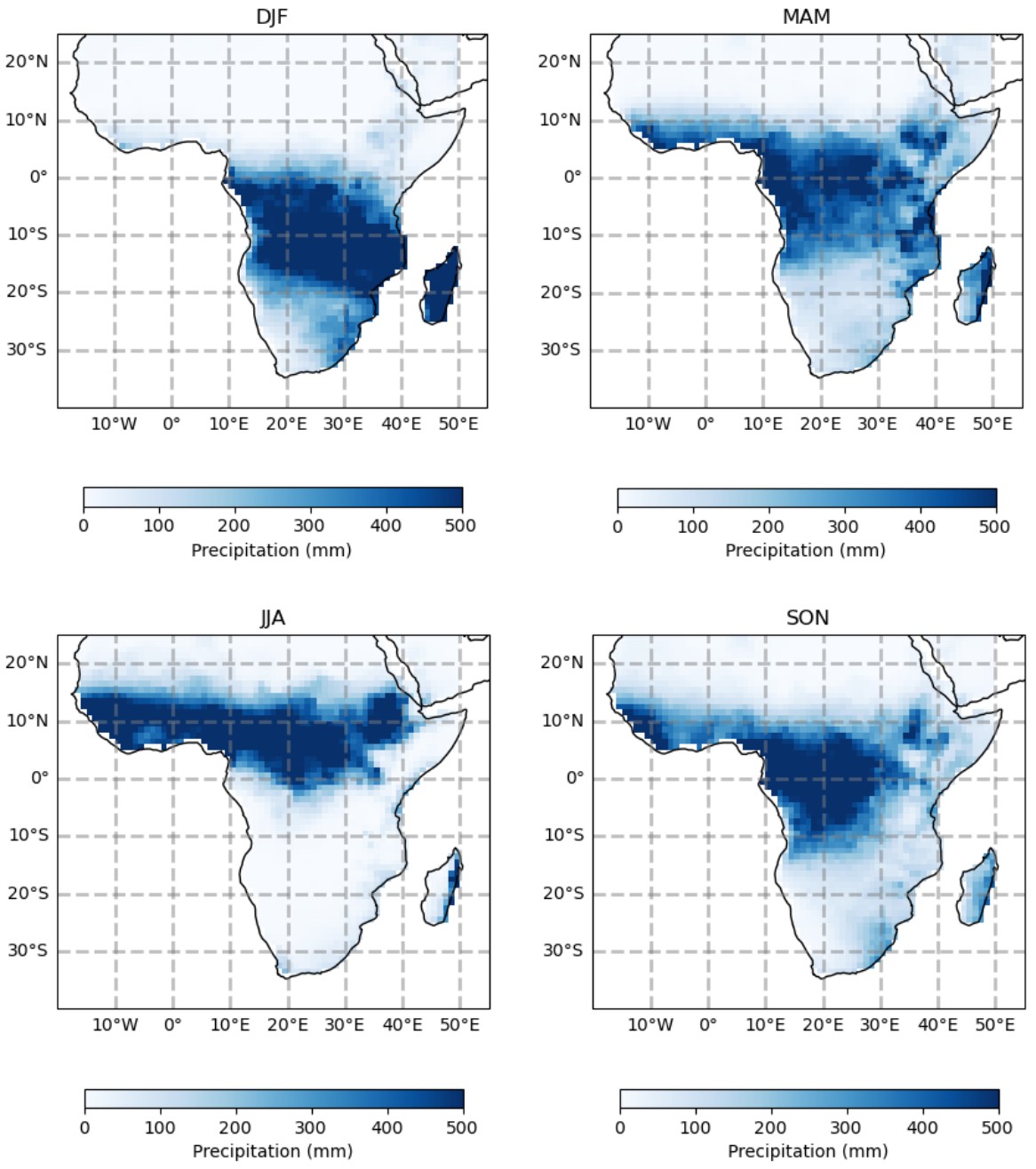

**Figure 8: Seasonal cycle in precipitation over Africa: precipitation accumulations for DJF (top left), MAM (top right), JJA (bottom left) and SON (bottom right)**

### 3.3.1    Case study 3: Data and methodology

The dataset chosen was the GPCC monthly gridded precipitation (a gridded dataset, based on rain gauge observations). For this study, the $1^0$ resolution 'Full Data Monthly Product' was used (Schneider et al. 2016). The weighting was carried out, using the Oceanic Nino Index (ONI) provided by NOAA (https://psl.noaa.gov/data/climateindices/list/).  SPI was calculated for each ensemble member precipitation time series, using the procedure outlined in Keyantash et al. 2023. The ROC-AUC scores are for detection of SPI < -0.75 (mild to moderate drought). The probability that SPI breaches the drought threshold are derived from the ensemble mean and standard deviation, assuming Gaussian behaviour. This choice is justified because SPI is constructed in such a way as to make it Gaussian, and an empirical approach tends to produce noisy results, especially for extremes. For the weighted ensembles, the weightings were derived from the ONI time series, with weights based on the difference between the ONI at the forecast initiation and the ONI for the climatological ensemble.

### 3.3.2    Case study 3: Results

Figure 9 shows example SPI forecasts initiated at the beginning of August, September and October 2015 for November SPI3. 2015 was a large El Nino event, resulting in high precipitation in East Africa and low precipitation in southern Africa. It can be seen that in August, when the forecast is based entirely on the climatological ensemble, as would be expected, the unweighted ensemble predicts climatology – i.e. zero anomalies. As additional data are included, the unweighted forecasts approach the observations, with most observed features evident by October. For this case (a large El Nino), comparison between the weighted and unweighted forecasts shows that the ONI weighting has a pronounced effect (Figure 9). Even in August, slight positive anomalies are predicted in East Africa and slight negative anomalies are predicted in southern Africa, in line with the expected El Nino teleconnection (Reason et al. 2017; Black et al. 2003). As the season progresses, the anomalies approach observations, with significantly positive anomalies evident in southern and eastern Africa at a 2-month lead.

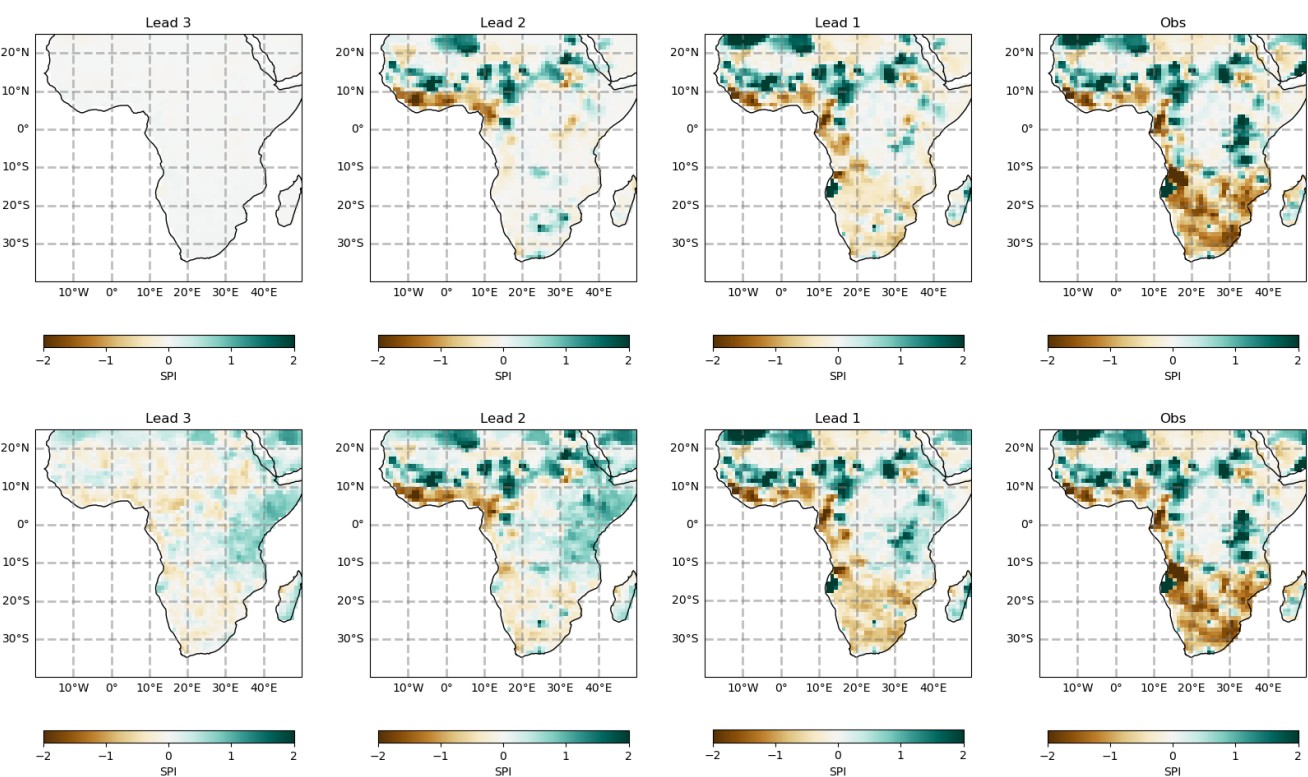

**Figure 9: Example forecast for 3-month SPI3 on November 15th 2015. From left to right: 3-month lead (forecast initiated on 15th August); 2-month lead (forecast initiated on 15th September); 1-month lead (forecast initiated on 15th October); Observed SPI3. The top row displays forecasts with no weighting applied to the ensemble. The bottom row displays forecasts, with the ensemble weighted using the Oceanic Nino Index (ONI).**

The improvement in skill as lead time reduces, suggested by the 2015 example, is consistent with the formal skill assessment shown in Figure 10, which shows the ROC-AUC for detection of SPI < -0.75 (mild/moderate drought). For the unweighted November forecasts, at a 3-month lead, the ROC-AUC is ~0.5 for all of Africa, indicating that the skill is no better than climatology. As observations are incorporated, the skill increases until it is >0.7 in most regions at a lead of one month. The weighting has the greatest effect at the longer lead times, with the greatest impact on East Africa. This is consistent with the well-known El Nino teleconnection with the East Africa October-December rainy season (Black et al. 2003).

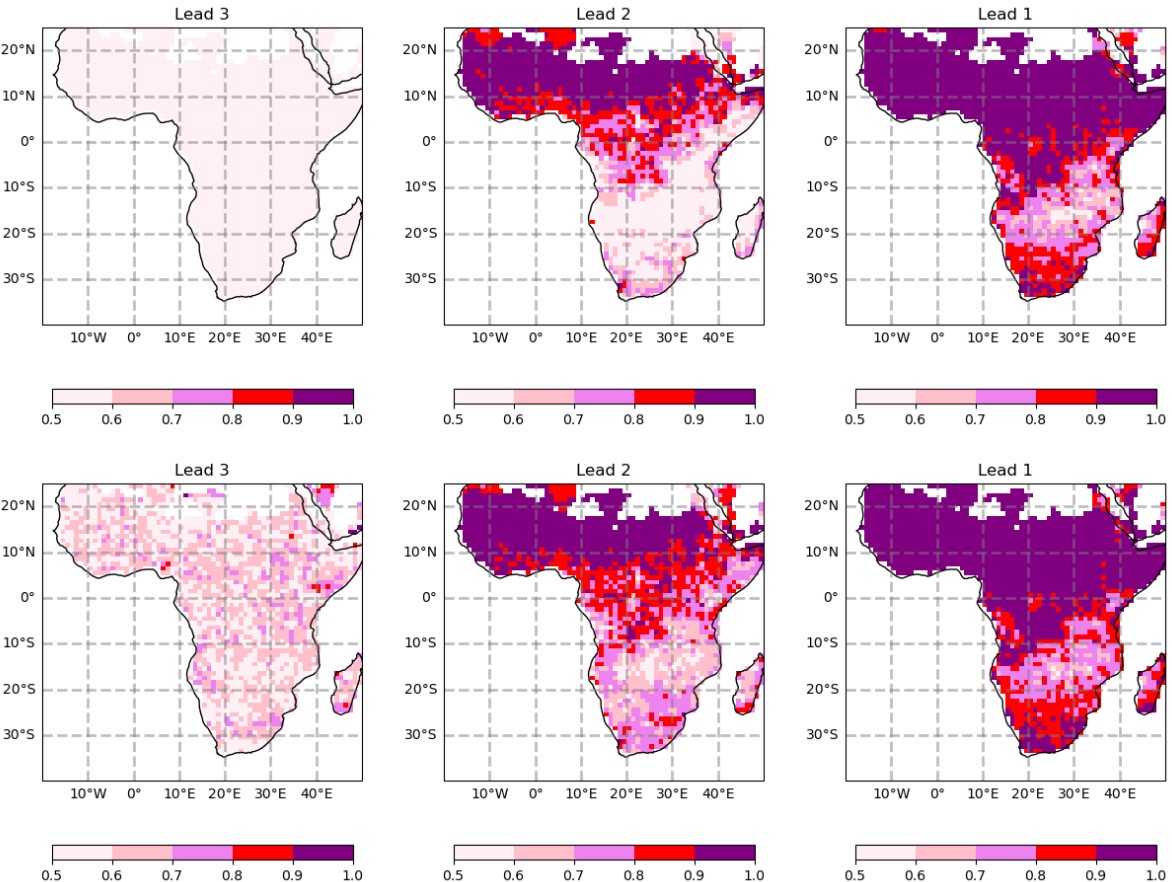

**Figure 10: ROC scores for 3-month SPI3 on November 15th. From left to right: 3-month lead (forecast initiated on 15th August); 2-month lead (forecast initiated on 15th September); 1-month lead (forecast initiated on 15th October); Observed SPI3. The top row displays forecasts with no weighting applied to the ensemble. The bottom row displays forecasts, with the ensemble weighted using the Oceanic Nino Index (ONI).**

## 4    Discussion and reflections

There are many excellent forecasting systems that contribute to early warning of climate-related hazard. Such systems are based on cutting edge numerical weather prediction models, and observations. These range from full dynamical systems run by meteorological agencies, such as the Met Office, ECMWF and the Bureau of Meteorology, to simple statistical regression models (e.g. Diro et al. 2008; Gissila et al. 2004). In recent years, there has been a proliferation of machine learning based algorithms, capable of emulating numerical forecasting systems or of inferring future conditions using historical data (see Ren et al. 2021 for a review). In parallel with forecast improvements, there have been significant developments in observation, and now that we are forty years into the meteorological satellite era, traditional ground-based observational networks are routinely supplemented by satellite estimates. Longstanding satellite datasets include the CHIRPS and TAMSAT precipitation data (Funk et al. 2015, Maidment et al. 2017), both of which provide data back to the early 1980s. Although generally less accurate than satellite and ground-based observations (for example, Lavers et al. 2022), meteorological reanalyses offer long

and consistent time series of a wide range of variables – some of which are not easily observed by more direct methods. Widely used examples include the NCEP and the ERA5 reanalyses (Kalnay et al. 2018, Hersbach et al. 2020).

So - what is the point of yet another ensemble forecasting system? From a technical perspective, General TAMSAT-ALERT is complementary to existing systems, which by and large focus either on observations or on forecasts. It is challenging to provide even qualitative assessments that take into account both the past and the future. For example, one can envisage a situation in which regional precipitation is forecast to be high, following a poor start to the rainy season. Does this mean that a seasonal meteorological drought is likely? Or is the forecast high precipitation sufficient to outweigh the current dry

conditions? General TAMSAT-ALERT provides a straightforward way to combine observations and forecasts into quantitative forecasts (Boult et al. 2020). Extending the example above, in some cases, meteorological forecasts may not be available, or they may have poor skill. In this situation, the method presented here enables decision-makers to judge the probability of seasonal drought, based purely on observations of the season so far (as with case study 3).

An additional issue addressed by General TAMSAT-ALERT is that of climate non-stationarity. The assumption of stationarity

is demonstrably false in a changing climate, and has been shown to have a detrimental effect on the skill of empirical statistical predictions (Salvi et al. 2016, Adeyemi et al. 2017). An advantage of the approach introduced in General TAMSAT-ALERT is that the weighting methodology accounts for non-stationarity without making assumptions about the structure of long term trends or variability. This means that the forecasts account for both natural decadal variability and anthropogenic climate change.

A further challenge for stakeholders is that the major forecasting organisations issue predictions and observations for a limited number of variables. Even the European Centre for Medium Range Weather Forecasting (ECMWF), which provides both forecasts and an extensive and continually updated meteorological reanalysis, only considers meteorological and land-surface variables that can be output directly from NWP models. Metrics of vegetation condition, such as NDVI, are not included, and could not be handled by the original TAMSAT-ALERT modelling framework. Lack of forecasts for particular variables may

make it difficult for stakeholders to accommodate forecasts into their operations – especially if their current monitoring protocols utilise variables, such as NDVI, that are not routinely predicted. The general approach to time series forecasting presented here provides a solution for such users.

The climatological approach central to the TAMSAT-ALERT methodology enables users to exploit publicly available meteorological forecasts and observations to improve skill (case studies 2 and 3 and Boult et al. 2020). Although incorporation

of skilful meteorological forecasts into TAMSAT-ALERT is undoubtedly useful (Boult et al. 2020), weighting with observed data is also potentially of value to decision makers. For example, El Nino can exacerbate the risks of extreme weather in many regions (for example Goddard and Gershunov 2020, Kay et al. 2022) and existing systems and warnings may report possible sectorial impacts of El Nino (for example Nobre et al. 2019, Kim et al. 2021). However, in practice, such reports tend to be semi-quantitative and reliant on expert judgement. The option to weight climatological ensembles with observations of climate

indices (weighting flag 2 in General TAMSAT-ALERT) supplements this information by providing quantitative assessments of how environmental variables are modulated by El Nino. The methodology implicitly accounts not only for varying teleconnection strength but also for the strength of the link between meteorological variables affected by El Nino and the metric being forecast. Figure 11 combines case study 2 (Pakistan NDVI) with case study 3 (African precipitation), comparing the effect of weighting with the ONI index during the strongest El Nino event in recent history. The results show that weighting

with ONI data has a strong effect on African precipitation forecasts, but that the predicted anomalies for Pakistan are uniformly weak - reflecting a weak link between El Nino and NDVI (although the predicted anomalies are stronger elsewhere in the region). The negative Pakistan result is of practical importance – especially given the anecdotal weight given to El Nino in the Pakistan meteorological and agricultural sectors (for example, Siyal et al. 2019). In the humanitarian sector there is often

discussion on taking action when an El Nino is imminent or ongoing, and yet in some cases the connection between El Nino and impact-relevant metrics, such as NDVI is weak. General TAMSAT-ALERT allows users to make their own judgements on the relevance of El Nino for their particular application. Allowing users to incorporate data on modes of variability is a significant extension of the original framework.

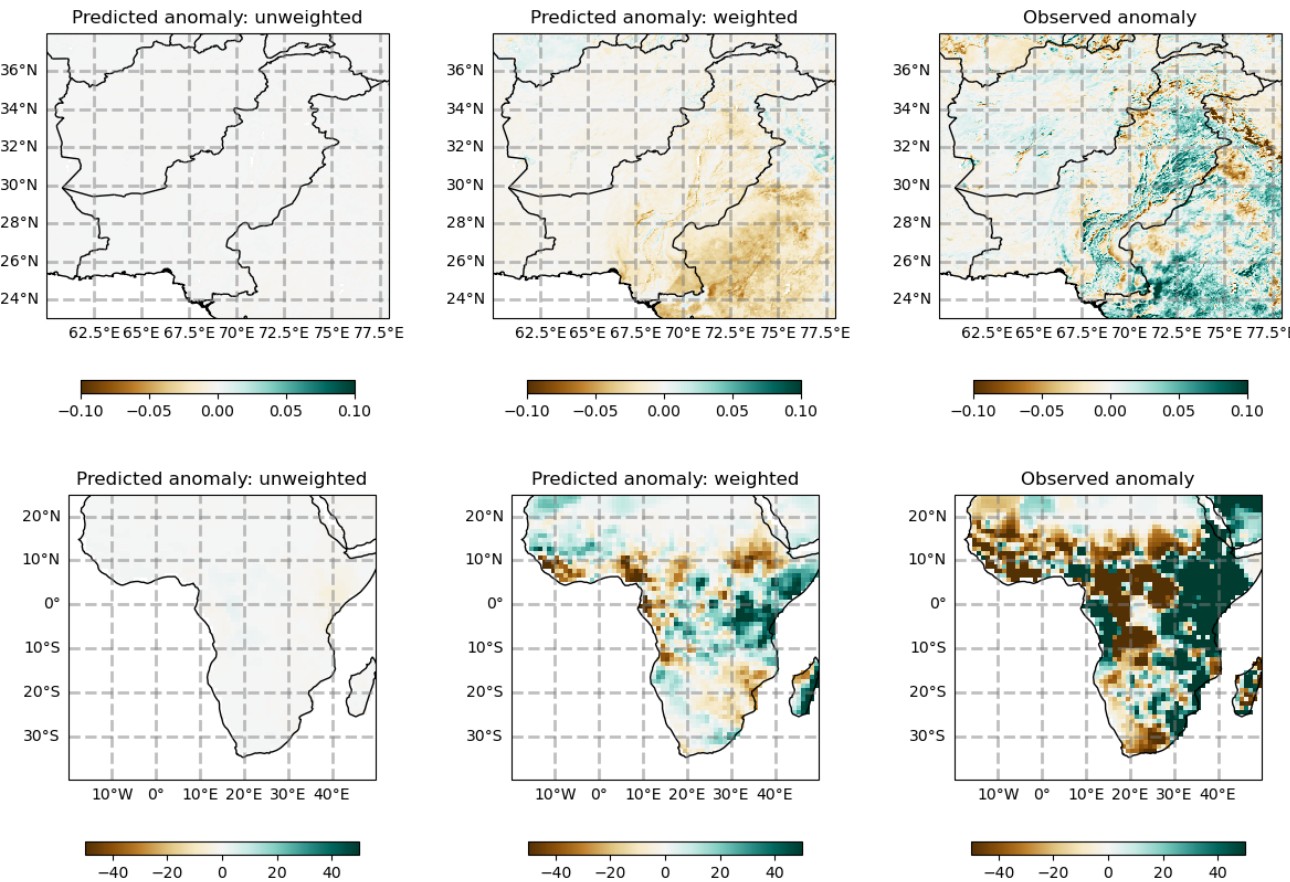

**Figure 11: Forecasts and observations of Pakistan (and surround region) seasonal mean NDVI anomaly (top) and Africa cumulative precipitation anomaly (bottom). Comparison between unweighted forecasts (left), weighted forecasts (middle) and observations (right). Forecasts are initiated on 1st August 1997 and target the September-November 1997 season. The weighting is based on the Oceanic Nino Index.**

The reliance of decision-makers on qualitative reports of El Nino highlights the limited uptake of forecasts for early action. This is in part because of the issue described above – namely that centrally issued meteorological forecasts may not target variables of interest. Further issues are that forecasts may not be available on the timetables specified in early action plans, and forecast metrics may not be presented in way that facilitates decision making. These issues can be addressed by some combination of collaboration, co-production, and direct production of forecasts by decision-makers (for example, Dasgupta et al. 2023). Our computationally light weight approach, and the public release of General TAMSAT-ALERT through the standard Python package indexing system facilitates all three solutions.

Connected to the discussion above is the somewhat nebulous notion of 'ownership' of the forecasting process. It is inevitable that forecasts are sometimes wrong. Loss of trust in forecasts is, however, not inevitable if decision-makers and end-users have a detailed understanding from the outset of the underlying principles, skill and subsequent limitations (Hirons et al. 2021, Gudoshava et al. 2022). Using a transparent method to generate ensemble forecasts and derive relevant metrics is an excellent way of building such understanding. Furthermore, end-users are more likely to persist with a system that they have built themselves – working with it over a number of years to design a decision-making process that accounts for error and uncertainty (Hirons et al. 2023).

## 5    Conclusions

In this paper, we have presented a computationally light weight and general method for ensemble forecasting. The key scientific innovations of the method are that it accounts for non-stationarity in time series and it exploits the predictability arising from persistence in some environmental variables, without the need to use an initialised numerical model. From a practical perspective, in comparison to numerical and machine learning forecasting methods, General TAMSAT-ALERT is easy to use and computationally light weight.

General TAMSAT-ALERT is designed to ingest multiple streams of meteorological forecast and environmental observational data. As such, the intention is not to replace existing forecasts and observations, but rather to provide a platform for transforming such data into actionable assessments. Over the next years, our aim is to demonstrate the use of the TAMSAT-ALERT methodology in a range of sectors, and to build the capacity of decision-makers to use our code to produce bespoke hazard assessments.

**Appendix A: A brief description of the TAMSAT-ALERT method**

**Overview**

TAMSAT-ALERT is a method of generating probabilistic forecasts of environmental variables, using observed time series and (optionally) meteorological forecasts or indices of large scale modes of climatic variability. As such, it enables users to combine multiple sources of data into probabilistic forecasts of environmental hazard. The way that TAMSAT-ALERT works

is illustrated by Figure A1.

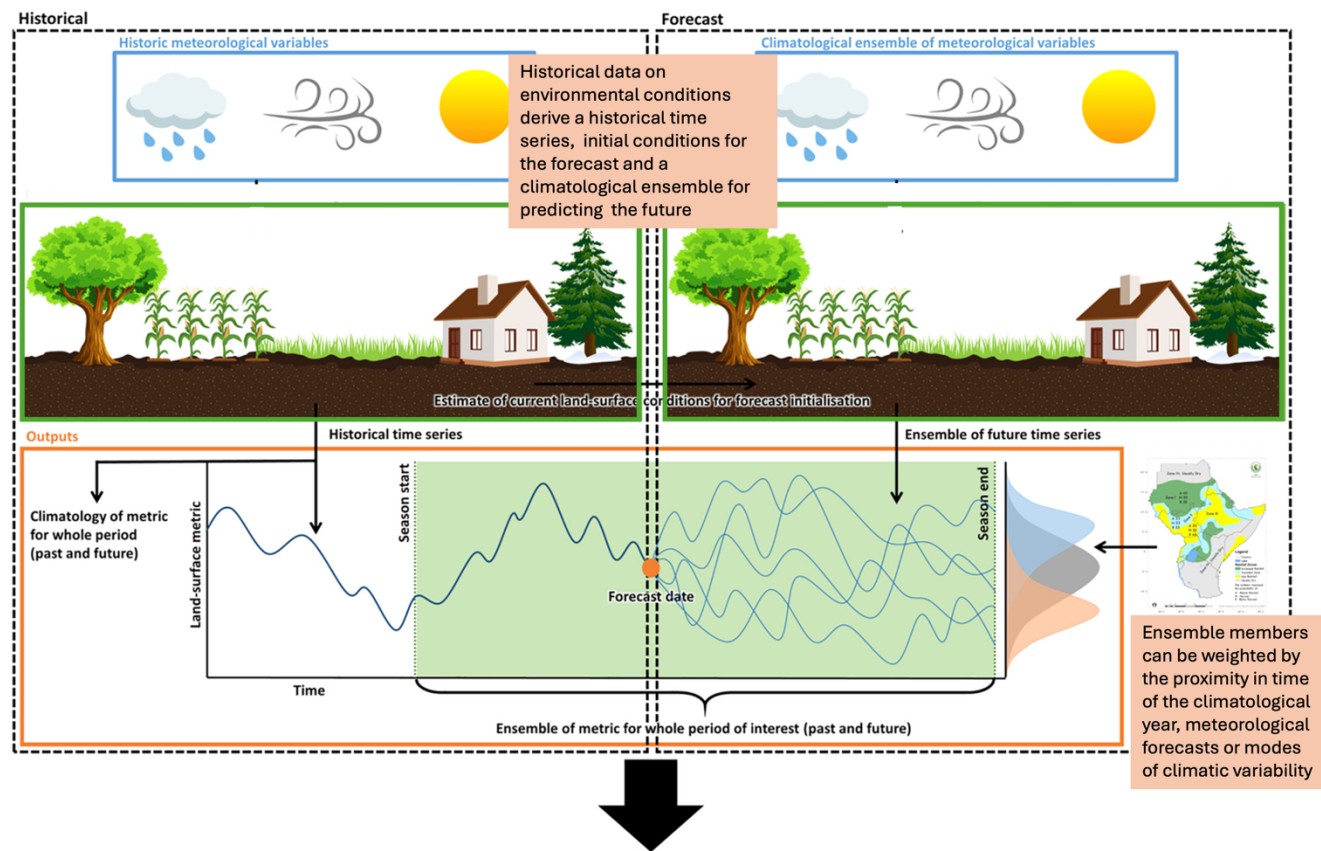

**Figure A1: Schematic of how the TAMSAT-ALERT method can be used to translate historical time series into probabilistic predictions**

It can be seen from Figure A1 that TAMSAT-ALERT forecasts uses historical time series to provide:

- Monitoring information on the part of the period of interest prior to the initiation date. In operational systems, forecasts are initiated as close to 'today' as possible, and the observational time series is used to monitor the progress of the season/period of interest so far

- Initial conditions for the forecasts

- A multi-year (climatological) ensemble. The climatological ensemble can be thought of as multiple realisations of
the future, with each possible future based on a series of events that have happened in the past

By using observational data from multiple years from the past, spliced with observational data from the period-of-interest so far, TAMSAT-ALERT thus generates an ensemble of time series for the whole period of interest.

**Weighting the ensemble**

In its default set up, TAMSAT-ALERT treats each ensemble member as equally likely. If additional information is available
about the probability of individual ensemble members, this information can be used to weight the ensemble. In practice, weighting the ensemble members is straightforward because each ensemble member is derived from a historical year. Thus, if

we can infer how likely conditions during a historical year are to occur during the period of interest, we can use this information to weight the ensemble. In this paper, and in previous studies, three methods of weighting the ensemble have been demonstrated:

Weighting method 1: By the proximity of the period of interest to the climatological ensemble year

Because many environmental variables show trends over time, ensemble members from years closer in time to the TAMSAT-ALERT forecast are more likely to accurately represent conditions in the future segment of the period of interest. In this weighting method, each ensemble member is therefore weighted based on how close its associated year is to the period of interest.

Weighting method 2: By the similarity of climatic modes of variability during the period of interest to modes of variability during the climatological ensemble year

In many parts of the world, the climate is greatly affected by patterns of variability, such as El Niño-Southern Oscillation (ENSO) and the North Atlantic Oscillation. This means that weather conditions during a given period are more likely to be similar to those in past years when these patterns were in a similar phase, rather than when they were in a different phase. For

example, in East Africa, rainfall is influenced by ENSO. If the period of interest occurs during an El Niño, the rainfall will be more like the rainfall during past El Niño periods than during La Niña periods. Weighting method 2 takes this into account by giving more importance to ensemble members for which a user-provided index of a variability pattern for the relevant climatological year is similar to the index on the forecast's start day. If the user-supplied index is poorly correlated with the variable being forecast, the effect of the weighting on the ensemble statistics will be minimal, while if it is strongly correlated,

the ensemble statistics will be significantly perturbed.

Weighting method 3: Using meteorological forecasts

Meteorological forecasts provide information on the likelihood that a specific weather variable, such as precipitation, will fall into one of three categories: below average (tercile 1), average (tercile 2), or above average (tercile 3)*. This information can be used to weight ensemble members. Each ensemble member is assigned to a tercile based on how the weather conditions

during the associated ensemble member year compare to the historical average. The weighting is then based on the forecasted probability for that tercile. For example, in East Africa, rainfall in November 1997 was above average (tercile 3). If we are using a November precipitation forecast for East Africa to weight the ensemble, the 1997 ensemble member will be assigned to tercile 3 and weighted according to the present day probability of rainfall in that tercile. This approach allows precipitation forecasts to weight TAMSAT-ALERT predictions for various variables (like NDVI). Continuing the example, similar to

weighting method 2, if the connection between rainfall and NDVI is weak, then using a precipitation forecast to weight the TAMSAT-ALERT prediction will have little impact on the ensemble statistics.

*other thresholds for categorising meteorological variables, such as quintiles and deciles are equally applicable

**Calculating the risk of an adverse event**

The risk of an adverse event can be calculated similarly to other ensemble predictions. For instance, the weighted ensemble

mean and standard deviation can determine the probability that NDVI or precipitation will breach a user-defined threshold. Another method involves using ensemble members to drive a model, such as a crop or land-surface model. In this approach, each ensemble member is used to run the model, the model outputs are combined into an ensemble, and the risk assessment is based on the weighted ensemble statistics.

**Appendix B: Additional information of the inbuilt method for weighting the climatological ensemble**

Within the General TAMSAT-ALERT, there are two inbuilt options for weighting the climatological ensemble. These are defined in the wrapper function by flags:

- weighting flag 0: no weighting applied
- weighting flag 1: weight the ensemble based on the proximity of the climatological ensemble year to the year in which the forecast is initiated

- weighting flag 2: weight the ensemble based on the similarity of some index at the point of forecast initiation to the index during the climatological year

The method of weighting is similar for methods 1 and 2. For weighting flag 1, the ensemble is weighted as follows:

$$w_i = e^{-0.001\left(S\,\Delta I\,\frac{T}{24}\right)^2}$$

where $w_i$ is the weighting for ensemble member $i$; $S$ is the weighting strength; $\Delta I$ is the difference between the initiation time

index and the climatological ensemble time index; $T$ is the dominant period of the data

For weighting flag 2, the ensemble is weighted as follows:

$$w_i = e^{-(S\,\Delta V)^2}$$

where $w_i$ is the weighting for ensemble member $i$; $S$ is the weighting strength; $\Delta V$ is the difference between the value of the climatic index at initiation and the climatic index for ensemble member $i$; $T$ is the dominant period of the data

The weighting strength for individual ensemble members for both methods is illustrated by Figure B1. The index used in the illustration is the Oceanic Nino Index (ONI) and the forecast was initiated in November 1997 (a strong El Nino).

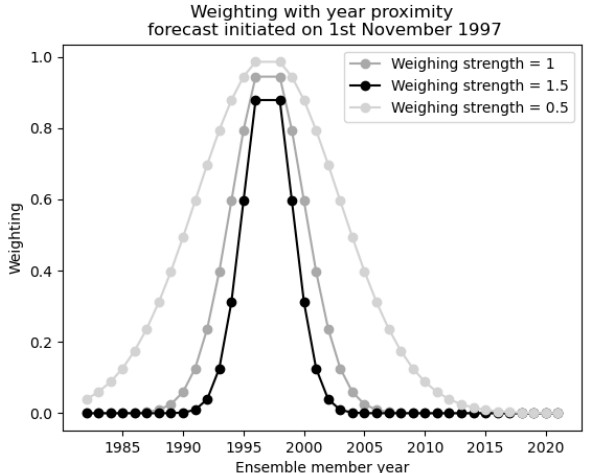 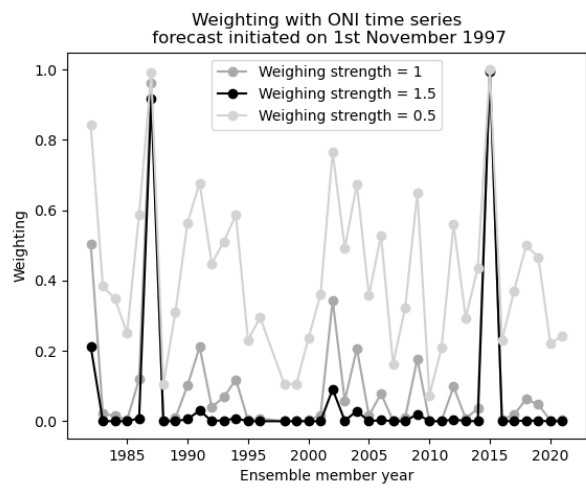

**Figure B1: An example of weights assigned to climatological ensemble members for a forecast initiated on 1st November 1997. Left panel: weighting by year proximity; right panel: weighting by the Oceanic Nino Index. Note that the year of the forecast (in this case 1997) is left out of the climatological ensemble**


**Appendix C: Additional information on the inbuilt method for determining the periodicity of the time series data**

For meteorological time series data, the dominant periodicity is almost always one year. It is therefore recommended that periodicity is specified by the user – accounting for the time resolution of the data. For example, for the monthly precipitation and temperature data in case studies 1 and 3, the user specified periodicity was 12, while for the twice-monthly NDVI data in case study 2, the user-specified periodicity was 24.

However, in order to make General TAMSAT-ALERT applicable to data without a known periodicity, a method for deriving periodicity is supplied within the python package. The methodology is illustrated in Figure C1and summarised below:

1. The input data is first transformed into a cleaner state by subtracting the mean square offset error. This method works by quantifying how different the signal is from itself offset by a given amount, effectively correcting for small variations in phase and removing noise from the data.

2. The data are linearly detrended

3. A Fourier transform is applied on the processed data

4. The maximum peak in the resulting spectrum is identified as the dominant periodicity

The algorithm described above is computationally intensive, so is applied to a subset of the gridded input – selected by regular sampling of the grid at user-specified intervals. It should be noted that the dominant periodicity is assumed to be constant in 560 space. If this is not the case, users are recommended to subset the data regionally before applying the TAMSAT-ALERT method.

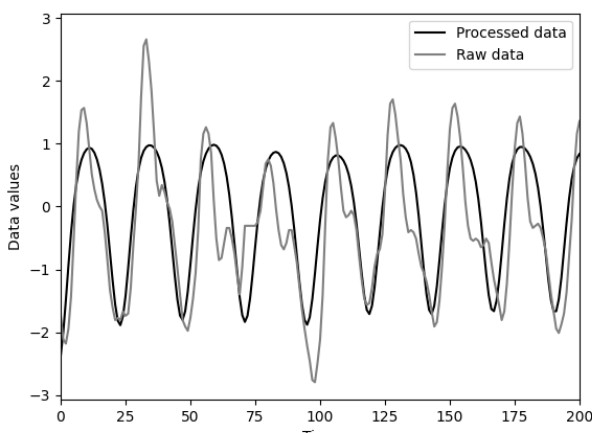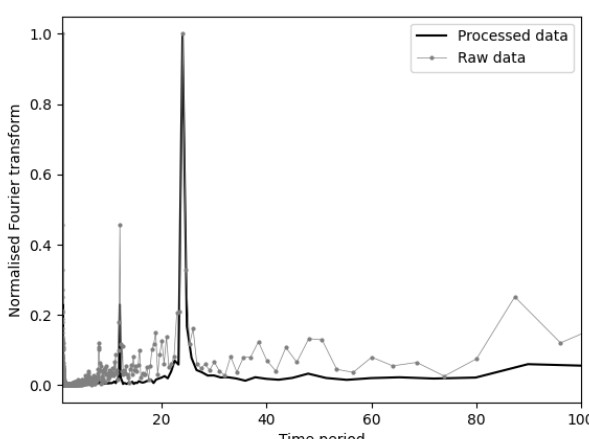

**Figure C1: An example of the dominant periodicity calculation method. Left panel: example time series of processed and raw data (steps 1 and 2 above) Right panel: Fourier transform of the raw and processed data (step 3)**


**Code availability**

All model code is open source and publicly available at https://pypi.org/project/general-tamsat-alert/ and https://github.com/brightlego/General_TAMSAT-ALERT. The version of the model evaluated at this paper is persistently archived at DOI: 10.5281/zenodo.10955490 A user guide to the code is included as supplementary materials.

The ROC score calculation code is available at https://pypi.org/project/fastroc/ and https://github.com/brightlego/fastroc

**Data availability**

All datasets used in this study are publicly available, via the sources given in Section 2. Convenience copies of the netcdf format files used to produce the plots are available at:

https://gws-access.jasmin.ac.uk/public/tamsat/tamsat_alert/gmd_paper/datasets.zip

**Author contributions**

EB led the writing of the manuscript and conducted the case studies. JE wrote the General TAMSAT-ALERT and fastroc code; devised the weighting and periodicity analysis methods and wrote Appendices B and C. RM processed the NDVI data, and leads the operational applications of TAMSAT-ALERT. All authors commented on the manuscript draft.

**Competing interests**

The authors declare no competing interests.

**Acknowledgements**

This work was supported by the National Centre for Atmospheric Science through the NERC National Capability International Programmes Award (NC/X006263/1).

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
