# Peer review of "A computationally light-weight model for ensemble forecasting of environmental hazard: General TAMSAT-ALERT v1.2.1"

_Geoscientific Model Development, 2024_

## Author Comment (AC1)

This paper documents a general method of producing forecasts by taking observations up to a point and combining with the historical distribution as a plausible ensemble of outcomes from that point. It is clearly a valuable tool (having already been used in several real-world contexts) and is worthy of publication, with the documentation allowing other researchers / users to deploy the method for themselves.

I have no hesitation recommending this for publication, barring a few minor comments listed below - mostly to improve clarity, although there is one suggestion that the years for the case studies should be changed such that the forecast method is "fair" and doesn't assume knowledge of future conditions from each forecast.

Thank you for your constructive comments. I have re-done the forecasts in the examples using only data from the past. I hope that an additional appendix, summarising the TAMSAT-ALERT method will help make the paper clearer,

Minor commnets

L30 "hazard(s)"

ok

L39 "drift in predictions" is this really a fundamental problem? Surely it is simply fixed with bias correction. OK bias correction is often not simple, but as written suggests that the problem is insurmountable. Perhaps a brief note here for completeness would be useful.

I have changed 'fundamental' to 'challenging' and have included an extra sentence explaining that this type of bias is difficult to correct for because it depends on the lead time.

L46 "seamless integration of past and future conditions facilitates in-season updates for slowly developing hazards". Is this really unique to the method? NWP/seasonal forecast systems do exactly this - accounting for past conditions (represented by the initial state, eg. initialised soil moisture capturing antecedant rainfall) and evolving into the future. This is easy to be updated in-season -  just run with a more recent initial state. Of course this doesn't stop it being a 'feature' of the GT-A method, but it would be clearer to really focus on it's uniquness (parsimonious and quick method of making a forecast for any variable by integrating current state with plausible future states, allowing testing of some simple assumptions around climate indices and non-stationarity)

I didn't intend to claim that this is a unique feature of TAMSAT-ALERT. However, it is surprisingly fiddly to combine past time series with future predictions that (a) do not have any 'jumps' on the initialisation date, (b) are not affected by model drift and (c) have consistent daily/monthly statistics throughout the time series. I have added in an extra sentence clarifying this.

L77 "continually" typo

Missing word now added (sorry for the oversight)

L82 "tThe" typo

Corrected

Figure 1 Nice figure. Would suggest simplifying the text in boxes as much as possible and making them active/imperative sentences rather than passive (i.e. put the verb at the front) e.g. "Define initialisation date and period of interest" (don't need to specifiy "user" since the box color tells us

that); "Calculate gridded difference time series". Also, is it possible to add the numbered steps from the previous stage on to this figure? Not quite sure it is, but it may be unecessarily confusing to have two parallel descriptions of the method which don't map on to each other.

Excellent suggestions. I have re-made the figure (see also response to R2)

L134 "incrementing forecasts". Not sure this is explained fully. I think you mean, that the historical data is transformed into anomalies relative to the initial state for that year, and then added on to the current state? e.g. if June 1981 is 23C and July 1981 is 24C, and June 2024 is 20C, then that 'member' produces a forecast of 21C for July 2024. Unpacking the method here or elsewhere would help clarify.

I have added a few lines on this to section 3.1.2 Case study 1: Data and methodology

L203 Setup 1 - essentially means that July FC is just the ensemble of values for historical Julys. This becomes obvious later but could be spelt out here.

This is clarified

L204 How are they weighted? I note the appendix giving details of the exponential weighitng, but this includes a coefficient. Which is used here?

The weighting coefficient is arbitrarily set to 1

L206 "Day of initialisation" is confusing since you're using monthly data in this example. Also missing a note to say you're making forecasts at a 1-month lead (June)

Clarified

Figure 2 missing subheadings ( (a) etc, but could also include the name of the set-up rather htan needing to refer to caption). Can also get rid of legend in 3/4 plots and use the space to make the plots bigger. Also, y axis units missing.

I think the referee meant Figure 3. The suggested corrections have been made, and I have used 2018 because it is a more striking example than the original choice.

Figure 4 Caption missing D

ok

Figure 5 units missing on top plot. Also please mention what the subnational borders represent (e.g. states ... although you could easily show this plot with just country borders -since the distinction of the admin1 regions is not relevant or discussed, and is not included on subsequent plots)

I have removed the sub-national borders and just included the country border for Pakistan. I have also made some other changes to the figure in response to R2.

Sec 3.1.1 Missing info on if forecasts were produced with weighitng, incremental option or both. Also now this case study uses gridded data, maybe worht pointing out that the method runs independently on gridpoints.

The forecasts were run with incrementation but no weighting. This is now clarified

Figure 6a Is it possible to achieve negative NDVI? If so please comment on what this represents.

It is possible to have negative NDVI. It usually signifies low/no vegetation cover and/or poor quality data. However, there were very few negative values in these plots and so I have adjusted the colour bar accordingly.

L273 The scores here are calculated now across all gridpoints for the single forecast?

Yes – that's right. Now clarified in the test.

L282 Did you test all the set-ups with Case study 2? If so do you find that the weighted years + perseistance brings minimal improvement beyond just using persistance?

I did do this, and there was a minimal (probably statistically insignificant) improvement, because there is no strong teleconnection or trend in the data. For the sake of brevity and maintaining focus I decided not to include these results in the paper. I see the purpose of the case studies is to illustrate the functionality of the General TAMSAT-ALERT method, rather than to provide any sort of comprehensive skill assessments.

Figure 7 please add an extra colour so the color changes match up with the labelled intervals

ok

L306 "(a)ccumulations"

corrected

L314 Deriving probabilities from ensemble mean and spread requires Gaussian assumption. Why not just count percentage of members below a threshold since it doesn't assume anything? Does it make a difference? If this is avoided deliberately please document the reason.

SPI really should be Gaussian (although the transformation is never perfect), and the use of empirical methods can be noisy especially for shorter time series and extremes. I have added this clarification into the text.

L325 "negative anomalies are also (incorrectly) forecast in southern Africa" - although, negative anomalies are correct for South Africa

Clarified

L425 I feel like it would help to discuss a little this weighting method in the main text. A few different weighting strengths are shown - are these discrete options for the user or can they set this to any value? It would probably also be helpful to give some guidance on which weighitng to choose (maybe the point is that it's up to the users to play around and decide what makes most sense to them - in which case that is also useful thing to say).

Additional text on the weighting has been added to the methodology section.

Figure A1: just looking at this it becomes clear that the forecasts in the case studies are made with knowledge of future conditions. This is not a fair forecasting test - really it should be an out-of-sample event, not including any 'future' information. I realise the paper is not primarily 'about' forecasting, but I would prefer to see the analysis reproduced taking this into account. For Case study 1 it could just be switched to July 2023 with presumably the same result. Also Case study 3 could also be switched to 2023 - which was also a very wet El Nino year in the region so should look similar. Case study 2 maybe more difficult, but is there a more recent drought that could be used (and, basing the forecast production on a period truncated to the year before that event?

For the years hindcast, the data for that particular year are removed from the climatological ensemble, so there should not be any substantial difference in skill. Regarding the case studies:

- For the CET case study, because 2021 was the last date within the file we were using, the example shown is outside the range of the climatological ensemble
- I have re-done the NDVI example for 2018, truncating the data at that point
- For the SPI, I truncated the data to 2015 and showing that El Nino

---

## Author Comment (AC2)

The manuscript presents General TAMSAT-ALERT which is an updated and extended version of an existing tool (TAMSAT-ALERT) to combine historical times series and climatological forecasts to obtain probabilistic forecasts that can consider climate variability and climate change by weighting members of the ensemble. It seems that General TAMSAT-ALERT is a very useful tool; it seems to be simple and easily applicable, and I think the presentation of General TAMSAT-ALERT deserves publication.

However, I feel that the manuscript is, in its current version, not very accessible to readers that are not deeply familiar with (meteorological) forecasting. Sentences like "… A key innovation is the option to increment variables from the initialisation date, enabling forecasts to account for persistence in time…" (Line 70) are not easy to understand outside the forecasting community. I had quite some problems in understanding the manuscript, and I could have provided a more detailed review if I had better understood the details. So, my main recommendation is to provide more information and to improve the understandability for readers outside the forecasting community.

Thank you for your constructive comments. I very much want this method to be accessible as possible, and I appreciate the feedback.

In addition to addressing your specific concerns, I have included an additional appendix explaining the methodology in a way that is, I hope, accessible to non-meteorologists. I hope that this additional methodological detail will complement the user guide and jupyter notebook, enabling the forecasting method to be useful to as wide a community as possible.

Specific comments:

The manuscript needs careful correction, as it contains a few typos. Some examples: Line 82: "… the …" instead of "… tThe …"; Line 87 and Figure 1 (bottom right box): bracket missing; Line 169: "… perfect forecast and the observed …" instead of "… perfect forecast, in the observed …"

Apologies – I have given the manuscript a thorough proof read.

Title, abstract and short summary: The authors should make it very clear that they mean climatological forecasting when they speak about forecasting. This is mentioned in the abstract but not in the title and the short summary. I was confused about that. For instance, the sentence in Line 120 (enabling users to derive forecasts directly from observations and reanalysis, without the need for the use of land-surface/crop models or NWP forecasts) should come much earlier.

Although the system is designed to be capable of combining meteorological forecasts with time series observations, it is not only capable of climatological forecasts, since not all weather-related hazards are climatological variables. For example, case study 2 is for NDVI. The confusion may be because of my use of the term 'climatological' for multi-year. I have clarified these points in the abstract, short summary and main text.

Line 40: " … A more fundamental problem is the drift in predictions … If model predictions were to be spliced directly onto historical observations, the drift would cause systematic bias in yield assessments, the magnitude of which would depend on the stage of the growing season at which the meteorological forecasts were initiated. The TAMSAT-ALERT approach addresses these issues by splicing together historical time series for the past with a climatology for the future …": Here, I am not sure whether I understand this argument correctly. Do you compare the TAMSAT-ALERT approach where you combine historical time series with forecasts (of today) to a situation where you combine historical time series with a model prediction that started in the past? In the latter approach: why would you not use historical data until today and then start the simulation?

See also response to Reviewer 1, and the additional text, which I think clarifies what I mean by this point. We do use historical until today and then start the simulation. The issue with using a model to generate the simulation is that the model output will exhibit a trend, as the model moves towards a state consistent with its internal physics. The TAMSAT-ALERT method is one possible approach to avoiding this problem.

Line 55: "… A strength of the methodology is that NWP output can be incorporated, even when forecasts are not available for the variable being assessed. In Kenya, for example, incorporation of skilful precipitation tercile forecast probabilities output by the ECMWF dynamical forecasting system improves the skill of NDVI and yield forecasts during the secondary rainy season …": Here, I would like to have more information on how the use one variable as proxy for another variables works.

I have included a new Appendix, which outlines the TAMSAT-ALERT approach and explains this.

Section 2.1 and Fig. 1: I propose to align the 9 steps in the text with Fig. 1. Currently, it is not obvious which step belongs to which box in Fig. 1 and why there are much more boxes than steps. It should also be shown in this figure (if possible) what the change/additions in comparison to TAMSAR-ALERT are.

Excellent suggestion (see also R1). I have re-made the figure.

Line 120: Do I understand correctly that the predecessor system TAMSAT-ALERT needed simulations models (land-surface/crop models or NWP forecasts) but General TAMSAT-ALERT does not need them? I am confused. Please provide a more comprehensive description of TAMSAT-ALERT.

This is explained in the new appendix

Line 166: in the introduction of the 2 skill scores, the predictand should be general, as only in the second case study NDVI is predicted.

ok

Figure 3: Please add (A), (B), (C), (D) to the subplots.

ok

Figure 5: This figure needs much more information to be understood. Please add the scale; what do the polygons mean? Where are the boundaries of your case study are? Maybe also show a few main cities, so that the reader can easily understand the figure. Why does the color bar end at 500 mm while in the text you write that there is rain up to 1000 mm?

I have remade the figure using a better colour scale and just including the Pakistan country outline to orientate readers.

Line 420: Please add a short Conclusions section.

I have included a short conclusions section.

**Citation**: https://doi.org/10.5194/gmd-2024-75-RC2